# Simulation of Friction Fault of Lightly Loaded Flywheel Bearing Cage and Its Fault Characteristics

**DOI:** 10.3390/s22218346

**Published:** 2022-10-31

**Authors:** Changrui Chen, Zhongmin Deng, Hong Wang, Tian He

**Affiliations:** 1School of Astronautics, Beihang University, Beijing 100191, China; 2Beijing Key Laboratory of Long-Life Technology of Precise Rotation and Transmission Mechanisms, Beijing Institute of Control Engineering, Beijing 100094, China; 3School of Transportation Science and Engineering, Beihang University, Beijing 100191, China

**Keywords:** flywheel, bearing fault, friction fault, load region factor, fault feature

## Abstract

Because of the operating environment and load, the main fault form of flywheel bearing is the friction fault between the cage and the rolling elements, which often lead to an increase in the friction torque of the bearing and even to the failure of the flywheel. However, due to the complex mechanism of the friction fault, the characteristic frequencies often used to indicate cage failure are not obvious, which makes it difficult to monitor and quantitatively judge such faults. Therefore, this paper studies the mechanism of the friction fault of the flywheel bearing cage and establishes its fault feature identification method. Firstly, the basic dynamic model of the bearing is established in this paper, and the friction between the cage and the rolling elements is simulated by the variable stiffness. The influence law of the bearing vibration response reveals the relationship between the periodic fluctuation of cage-rolling element friction failure and the bearing load. After analyzing the envelope spectrum of the vibration data, it was found that when a friction fault occurred between the cage and the rolling element, the rotation frequency component of the cage modulated the rotational frequency component of the rolling element, that is, the side frequency components appeared on both sides of the characteristic frequency of the rolling element (with the characteristic frequency of the cage as the interval). In addition, the modulation frequency components of the cage and rolling element changed with the severity of the fault. Then, a modulation sideband ratio method based on envelope spectrum was proposed to qualitatively diagnose the severity of the cage-rolling element friction faults. Finally, the effectiveness of the presented method was verified by experiments.

## 1. Introduction

A bearing subassembly is a core component of a flywheel system, and it considerably influences the performance of a spacecraft, such as a satellite. Hence, flywheel bearings should be thoroughly monitored. As the demand for rotational speed, rotation accuracy, and load increases, the cage friction fault becomes more influential in determining bearing performance [1]. The cage is a highly problematic part of the flywheel bearing, which can cause bearing failure. However, the cage is still difficult to monitor and diagnose for faults because its friction fault mechanism is complicated while the fault characteristics are weak. Therefore, it is crucial to establish a model for analyzing the dynamic response of the cage friction. Meanwhile, it is theoretically significant to extract the required fault characteristics for identifying flywheel bearing faults.

Vibration-based methods are currently used for monitoring flywheel bearings [2]. Defective bearing components affect machine stability and performance; thus, the abnormal vibration signals produced by defective components are suitable for determining bearing conditions [3].

The extraction of significant features is essential for the efficient fault diagnosis and prognosis of rolling element bearing [4]. Zhang et al. [5] proposed a periodic low rank dynamic mode decomposition algorithm to extract features. Cui et al. [6] used the machine learning to extract the bearing fault features. Wang et al. [7] proposed a robust fault characteristic extraction approach based on the time-frequency analysis. Cai et al. [8] proposed the use of a rule-based algorithm and Bayesian networks (BNs) or Back Propagation neural networks. Cai et al. [9] contributed a RUL re-prediction method based on the Wiener process. Kong et al. [10] proposed a sensor placement methodology of the hydraulic control system. Cai et al. [11] proposed the use of a rule-based algorithm and Bayesian networks (BNs) or Back Propagation neural networks. These studies show that capturing the specific fault characteristics of bearings is conducive to a better diagnosis, and feature extraction is one of the current research focuses.

As balls are vital bearing components, ball misalignment, shaft slopes, surface roughness, and the high extent of waviness and inclusion may cause bearing fatigue, cracks, galling, spalling, pitting, and slippage [12,13]. Hence, several fault models have been proposed. Guo et al. [14] proposed a bearing fault diagnosis method based on the speed signal.McFadden and Smith [15] proposed a multi-fault model using a series of cyclic impact forces to determine the failure dynamic. Tandon and Choudhury [16] examined bearing faults by considering three shock functions. Niu et al. [17] modeled the localized surface defects of raceways as quadratic curves and conducted simulations and experiments to analyze the motion characteristics of bearings. In this approach, the authors considered changes to the Hertzian contact coefficients and contact force directions. Subsequently, they [18] examined a calculation method for the ball passing frequency on raceways with surface defects. Zhao et al. [19] analyzed the vibration features of a pitting fault gear system considering eccentricity and friction. Liu et al. [20] proposed a defect simulation method and established a relationship between the defect size and the impulse response. Cao et al. [21] modeled the shape of the transient impact forces of the bearing components’ local defects as a triangular form. Moreover, local defects are predominantly in the form of cracks, pits, and spalls on the component surface [22]. Bartłomiej et al. [23] presented a two-degrees of freedom (2-DOF) dimensionless mathematical model. These studies demonstrate that bearing faults mainly result from balls or raceways.

However, compared with ball-to-race contact loads, all cage forces, such as lubricant traction and frictional forces, are light, and the cage rotation is typically slower than the ring speed [24,25]. Thus, the cage fault modes and dynamic characteristics differ considerably from other bearing components in the light load case.

The cage is another highly problematic bearing component [26]. The interaction forces between the cage and rolling elements considerably influence the dynamic characteristics of the cage [27]. Several dynamic fault models have been proposed for studying the bearing cage characteristics. Bearing faults may result from the effects of moment load [28], flexibility [29], unbalance [1,30], friction [31], and clearance [32]. In the aspect of cage instability, He et al. [33] investigated the bearing cage under normal, rubbing, and uneven lubrication conditions.To clarify the occurrence of cage whirl motions, Kingsbury [34] proposed a cage whirl motion model and analyzed the influencing factors of the cage whirl radius [35]. Liu et al. [36] proposed a skidding dynamics model to reveal the skidding characteristics of the cage. In the model, the cage is discretized into several segments by springs connecting the adjacent segments. Shi et al. [37] discussed the effect of cage crack initiation and propagation on bearing performance. Gao et al. [38] discussed the relationship between the overall bearing skidding degree and the whirl characteristics of the cage.

The friction mechanism and diagnosis methods of the cage friction problem are discussed in the literature [39,40]. Pederson et al. [41] examined the effects of friction between the cage pocket and the rolling element on cage stability. Boesiger [42] found that the unstable whirl motion exits when the friction coefficient between the ball and cage changes. Kannel and Snediker [43] suggested that momentary changes in torque and force can result in the cage instability problem, which is a factor of cage friction. Chang et al. [44] reported that abrasion, rather than fatigue, might be a more prominent failure mode of RBEs under oil–air lubrication environments. Gupta [1] simulated cage wear considering the frictional characteristics of the cage. The author noted that the cage wear rate was correlated with the level of cage unbalance. Zhang et al. [45] proposed a dynamic wear simulation model to quantitatively analyze the bearing wear properties, and the wear volume was determined by the decrease in the ball diameter. These studies show that cage friction is correlated with cage instability. Du et al. [46] established an oil film flow model and explored the influence of the fluid characteristics of the lubricating oil film on bearing wear. Madar et al. [47] presented a study on the damage severity estimations of spalled bearings.

However, the fault information obtained by changing the loading environment or running conditions of a bearing is insufficient to describe the health condition of the bearing. Moreover, only a few studies have investigated the vibrations caused by the cage-ball friction fault. In addition, previous studies have mainly focused on the synchronous and nonsynchronous vibrations caused by friction faults; the cage friction fault features have been rarely studied from an actual bearing system perspective. Wang et al. [48] noted that rolling element bearings are the core support components and play a decisive role in the performance, operation reliability, and service life of the flywheel. The cage friction fault is the major fault type of flywheel bearings in the inchoate condition monitoring program. The structure and load-carrying environment of the flywheel bearing are different from those of the classical bearing system. For example, the fault mechanism of the cage friction problem of flywheel bearings is yet unclear, severely limiting the monitoring and fault diagnosis of space bearings. Therefore, this paper establishes an analytical model for the cage friction to determine the effect of the friction fault and explore the dynamic response features considering the variation in the load region. This study provides an important reference for the monitoring and fault diagnosis of flywheel bearings.

The remainder of the paper is organized as follows: in Section 2, the 2D model of the cage-ball friction fault for lightly loaded bearings is established; in Section 3, based on the proposed simulation model, the features of the cage friction fault are analyzed under different key parameters; in Section 4, the features of the cage friction fault were confirmed by an experiment. Finally, the conclusion of this paper is summarized in Section 5.

## 2. Dynamic Modeling of Cage–Ball Friction Fault for Lightly Loaded Bearings

### 2.1. Dynamic Model of Rolling Element Bearings

The actual movement of bearing components is highly complicated as many factors cause excessive friction between the cage and rolling elements. Hence, a dynamic simulation model is necessary for investigating the dynamic response characteristics of friction faults. During the health monitoring of bearings, it is easier to obtain the vibration signals emitted from the whole bearing than it is for the cage. Therefore, it can be assumed for simulation purposes that the cage and rolling elements are excessively rubbed together; thus, a dynamic bearing model that ignores the cage motion can be established. A simulation using such models can provide a theoretical reference for diagnosing actual bearing friction faults. Current dynamic simulation models of this kind are mainly based on Fukata’s work, which is a (two-degrees of freedom) 2-DOF model based on the Hertzian theory [49]. Subsequently, Feng et al. [50] improved upon this work by considering the effect of the bearing pedestal. By combining the above models and the motion characteristics of flywheel bearings, we established a 2-DOF model in this study, as shown in Figure 1 and Figure 2, and the model was used to analyze the cage–ball friction fault.

The flywheel bearing is different from ground bearings; namely, the operation of flywheel bearing is the outer race. In the model, the inner race is assumed to be stationary, while the outer race is regarded as the rotational unit, as shown in Figure 1. The outer race has 2-DOF in the horizontal and vertical directions; the rolling element–raceway contact is considered as a spring–damp system, and contact force is determined according to the Hertzian contact theory.

In this simplified bearing model, the outer race is driven at a constant velocity of $\omega_{n}$. In Figure 2, m, K, and c are the outer race mass, stiffness, and damping, respectively. The simplification of the bearing motion is based on the following:(1)The rolling element–raceway contact is regarded as the surface contact according to the Hertzian contact theory.(2)The rolling element–raceway slip is ignored.(3)The bearing components are rigid, except those at the contact area; thus, the rigid body rule is applied to the rolling elements in the non-contact regions.


The outer race displacements are x and y. The contact deformation $\delta_{j}$ of the jth ball with clearance ε can be expressed as:(1)$$\delta_{j}=-x\cos\theta_{j}-y\sin\theta_{j}-\varepsilon$$

The angular position of the jth ball $\theta_{j}$ is a function of the cage angular speed $\omega_{c}$, N is the number of balls, *t* is the time elapsed, and $\theta_{0}$ is the initial reference position
(2)$$\theta_{j}=\left(2\pi /N\right)\left(j-1\right)+\omega_{c}t+\theta_{0}$$
where $\omega_{c}=0.5\omega_{n}\times \left(1+D_{b}/D\right)$, $\omega_{n}$ is the angular speed of the outer race, $D_{b}$ is the ball diameter, and D is the bearing pitch diameter.

According to the Hertzian contact deformation theory, the ball force can be expressed as:(3)$$\begin{array}{l} F_{x}=K\sum_{j=1}^{N}\delta_{j}{}^{3/2}\cos\theta_{j}\cdot h\left(\delta_{j}\right)\\ F_{y}=K\sum_{j=1}^{N}\delta_{j}{}^{3/2}\sin\theta_{j}\cdot h\left(\delta_{j}\right) \end{array}$$
where K is the load-deflection factor of the balls and depends on the contact geometry and the elastic contact of the material [51]; h(⋅) is the Heaviside function whenever the independent variable of the function is greater than 0. Here, the function value is 1.

Therefore, the equations of motion for the 2-DOF bearing model can be expressed as:(4)$$\begin{array}{l} m\ddot{x}+c\dot{x}+F_{x}=mg,\\ m\ddot{y}+c\dot{y}+F_{y}=0 \end{array}$$
where m is the outer race mass, g is acceleration due to gravity, and c is the damping coefficient.

### 2.2. Mechanics Model of Friction Fault

For high-speed rolling bearings, the cage–ball friction phenomenon can considerably influence the bearing stability. Houpert et al. [52] found that cage–ball friction can cause bearings to produce an additional braking effect. Dumitru et al. [53] noted that the cage–ball friction is a crucial friction source in extremely low axial force conditions. Under high-speed and low-load conditions, rolling bearings often experience skidding, which can cause friction and incipient failure to the bearing ring and rolling element surfaces [54]. Jiang et al. [55] proposed a time-frequency spectral amplitude modulation method. Deng et al. [56] proposed an improved dynamic model of ball bearings with cage whirl motion and elastohydrodynamic lubrication.

When excessive friction occurs between the cage and rolling elements, the friction between both components increases instantly because of the rolling element extrusion effect on the cage pockets. The change in the friction force acting on the rolling elements is denoted by δf, as illustrated in Figure 3. The additional normal contact force ΔF and the moment $T_{\Delta f}$ are the obvious effect of δf; ΔF is the reaction force between the faulty balls and raceway, which can affect the spring force amplitude of the ball. $T_{\Delta f}$ is typically around the ball centroid and can change the rotation speed, leading to the braking effect. The revolution speed of the fault balls is either fast or slow because of the braking effect, which may intensify the impact friction between the cage and rolling elements, thereby increasing the likelihood of friction faults occurring. Therefore, $T_{\Delta f}$ was not considered in this study.

According to the assumptions of the rigid ball, the variation law of δf can be replaced equivalently by the reaction force ΔF, and then the cage–ball excessive friction problem can be addressed. The degree of hold-down between the fault balls and raceway varies according to the azimuth angle of the balls; thus, the rolling element extrusion effect on the cage pockets differs, and the ΔF amplitude varies with the azimuth angle of the fault balls. To account for the association between ΔF and the azimuth angle, we proposed a variable stiffness method to simulate the effects of the cage–ball friction fault: the load-deflection factor K of the fault balls varies with time and is expressed by the block-pulse function. The rolling element–raceway contact force F calculated by the variable stiffness method changes simultaneously with the degree of hold-down between the fault balls and raceway. Therefore, the effect of the cage–ball friction on the nonlinear vibration response can be determined by indirectly varying K. The variable stiffness method is detailed in the next section.

Meanwhile, the centrifugation trend of the fault balls is considerably suppressed as the cage friction fault occurs. Thus, the entire centrifugal force acting on the outer race can become unbalanced, as illustrated in Figure 4. At this time, the heavy out-of-balance force $F^{ce}$ occurs because of the high-speed rotation of the outer race.

The direction of $F^{ce}$ is along the opposite side of the fault balls, as illustrated in Figure 5, and the association in terms of direction is achieved by adjusting the initial reference position $\theta_{0}$. The variation period of $F^{ce}$ can be divided into two categories. When the bearings are in a light load state and because $F^{ce}$ induces the load region variation, the variation period is equal to the cage evolution period. However, the variation period does not exist when a constant force exists that can contend with the effect of $F^{ce}$. Owing to certain factors such as lubrication, rotational motion, and collision impact, the amplitude relationship between the out-of-balance force $F^{ce}$ and the reaction force ΔF is nonlinear, and it is difficult to express them mathematically. As an upper limit exits for the effect of the hold-down between the fault balls and raceway, the reference range (<constant) is used to examine the influence of the out-of-balance force amplitude on the bearing vibration. Once cage friction faults occur, a value must be obtained in the range that can represent the out-of-balance degree of the outer race. Thus, the friction fault is intensified as the value increases.

#### 2.2.1. Variable Stiffness Method of Simulating Friction Faults

The most extensive mathematical model for the local defects of bearings typically treats the defects as a consequent series of impulses while the bearing runs at a constant frequency [57,58]. With the bearing rotation, these impulses occur periodically at a frequency that is uniquely determined by the defect location. As stated previously, the reaction force ΔF is excited because the cage–ball friction increases, causing the outer race to generate these impulses. The impulse period depends on the geometric structure of the bearing and the outer race rotating speed. As ΔF and the azimuth angle are correlated, it will be inappropriate to provide a constant value for ΔF. Therefore, we proposed the variable stiffness method to simulate the effects of the cage friction fault: the amplitude of the load-deflection factor K is made a time-varying factor according to the fault balls, and the time-varying patterns described by the block-pulse function are expressed as:(5)$$\begin{array}{l} K^{*}=K+\Delta K\\ \Delta K=\xi \cdot e\left(t\right)\cdot K \end{array}$$
where $K^{*}$ is the load-deflection factor of the fault balls, and ξ is the stiffness change factor and extrusion extent between the cage and balls. The friction force changes in Δf are reflected through the amount and sign of ξ. The time-varying behavior e(⋅) of the fault source can be expressed by the block-pulse function as:(6)$$e\left(t\right)=\sum_{n=0,1,\ldots }\phi \left(t-nT_{0}\right)$$
where $T_{0}$ is the impulse repetition cycle and is considered to be the rotation period of the balls. The unit block-pulse is expressed as:(7)$$\phi \left(t\right)=\{\begin{array}{cc} 1 & 0\leq t\leq \tau \\ 0 & t>\tau \end{array}$$
where τ is the duration.

#### 2.2.2. Unbalance Effect of Outer Race

Owing to the cage–ball excessive friction, the centrifugal force of the balls acting on the outer race completely loses its balance, producing a severe out-of-balance force resulting from the high-speed rotation of the outer race. According to the motion characteristics of the bearing, the out-of-balance force is an alternate force, which can be denoted by the harmonic function and expressed as:(8)$$\{\begin{array}{c} F_{x}^{ce}=f_{ce}\cos\left(\omega_{c}t+\varphi \right)\\ F_{y}^{ce}=f_{ce}\sin\left(\omega_{c}t-\varphi \right) \end{array}$$
where φ is the phase difference, and $f_{ce}$ is the amplitude coefficient of the out-of-balance force $F^{ce}$ ($f_{ce}\in \left[c_{1},c_{2}\right]$). The value of $f_{ce}$ is uniquely determined by the reaction force ΔF, but the relationship between both quantities is yet unclear. The value of $f_{ce}$ under certain values of ΔF can be large or small. If the value is large, the friction fault is likely to be intense, and no unbalance effect exists for the outer race when $f_{ce}$ is zero. We selected a finite number of $f_{ce}$ to examine the nonlinear vibration of the cage friction fault under different severity levels.

Therefore, the equations of motion for the outer race can be expressed as:(9)$$\begin{array}{l} m\ddot{x}+c\dot{x}+F_{x}=mg+F_{x}^{ce},\\ m\ddot{y}+c\dot{y}+F_{y}=F_{y}^{ce} \end{array}$$

## 3. Friction Fault Analysis Based on the Proposed Model

### 3.1. Simulation Setup

Since Equation (9) comprises strong nonlinear factors, the Newmark-β method was applied to solve the system’s responses. The rolling element of the MB-ER-16K bearing used for the simulation was a deep groove bearing whose geometry size was similar to that of flywheel bearings. The bearing and fault parameters are listed in Table 1. The external load acting on the bearings was ignored, and for the outer race speed of 3600 rpm, the time increment was set to $1\times 10^{-6}$ s.

The periodic variation in the assembly stiffness that occurs as the cage rotates was the most influential cause of the rolling bearing vibration, which is the varying compliance (VC) vibration [60]. The VC vibration characteristics were used to verify the accuracy of the solution. The frequencies of the MB-ER-16K deep groove bearing are listed in Table 2. The vibration responses were calculated under normal bearing conditions; Figure 6 and Figure 7 illustrate the simulation results. Figure 6 illustrates the vibration acceleration in the vertical direction, and Figure 7 illustrates the envelope spectrum of the acceleration signals.

Figure 6 and Figure 7 show that the frequency components were of the VC characteristic frequencies as well as its dividing frequency components, which was highly similar to a principle from a previous work [61]. Therefore, key parameters, such as integration step, validated the numerical stability of the solution process.

### 3.2. Characteristics of the Cage-Ball Friction Fault under Different Radial Loads

We assumed that the stiffness change factor ξ was 0.1 while the fault balls were numbered two and four. The vibration responses were calculated in two loading environments: light and heavy loads. For the light load case, the reference range of $f_{ce}$ due to the unbalance of the outer race ranged from 1 N to 100 N. Figure 8 and Figure 9 illustrate the vertical acceleration time-domain waveforms and envelope spectrum for a 100 N amplitude coefficient $f_{ce}$. For the heavy load case, a constant 100 N force was applied on the outer race in the vertical and horizontal directions while $f_{ce}$ was set to 0 N. Figure 10 and Figure 11 show the vertical acceleration time-domain waveforms and envelope spectrum at a constant force of 100 N. We drew the following conclusions from the simulation results:

(1) The envelope spectra clarify the frequency of the cage and ball as well as their dividing frequency components, which result from the friction fault. Some modulation sidebands also take the cage frequency as the interval. The amplitude of the second-order sidebands of $f_{c}$ is significantly higher than that of the first-order for the light load case; the result of the heavy load case is the opposite. The sideband regularities around the harmonic components, such as 1X, 2X, and 3X of the ball spin frequency, are similar. Therefore, the characteristic spectra lines caused by the cage-ball friction are highly sensitive to the loading environment. As the external load is insufficient to offset the unbalancing effect of the outer race, the load region changes, causing the envelope spectra pattern.

(2) The distribution regularity of the amplitude of three sideband families differs, although it has similar modulation sideband phenomena around the first three orders of ball spin frequency. Hence, the influence of the outer race unbalance on the sideband amplitude is only a qualitative conclusion.

### 3.3. Analysis of Fault Parameter

#### 3.3.1. Influence of Stiffness Change Factor

To study the influence of the cage-ball friction fault on the bearing vibration, we reduced the stiffness change factor to ±0.8ξ and gradually increased it to ±1.5ξ in small ranges. The amplitude coefficient $f_{ce}$ was 1 N and 100 N. The characteristics of the vertical acceleration signal were then analyzed. Figure 12 and Figure 13 illustrate the vibration acceleration RMS values under different values of $f_{ce}$ (100 N, 1 N). It could be seen that the stiffness change factor ξ substantially affected the vibration energy at a certain value of $f_{ce}$. This effect occurred because ξ represents the friction fault severity, and the larger friction fault can stimulate a much larger vibration energy in the bearing system. Figure 14 and Figure 15 illustrate the peak value of the envelope spectrum curve at the sideband modulation family of the first-order ball spin frequency $f_{b}$. These figures show that the modulation sideband phenomenon under different values of ξ (±0.8ξ, ±1.5ξ) was mostly unchanged in ξ. Moreover, the acceleration RMS value and the amplitude of the characteristic frequencies were close regardless of the impact between the cage and balls in the positive direction (+) or negative direction (−); the energy from the cage-ball friction fault was transferred through the balls to the outer race in the variable stiffness method, which lacked any directionality.

#### 3.3.2. Analysis of Unbalance Force Influence on Load Region

When the bearings are running, the load region is the angular extent of the raceway contact deformation. The load distribution highly influences the vibration and dynamic characteristics [61], and the radial load has a complex relationship with the load region. Owing to the unbalancing effect of the outer race, the load region varies in the light load case; however, the load region variation is rarely considered in the study of the cage friction fault. We investigated the influence of the unbalance force on the load region in the light load case by considering the rolling element–raceway contact force. The vibration responses of the cage friction fault under a finite number of unbalance values were simulated. The unbalance effect was achieved in the simulation by changing the amplitude coefficient $f_{ce}$ of the out-of-balance force. Figure 16 shows the time domain waveform of the rolling element–raceway contact force in the heavy load case, and we can see that there is no amplitude fluctuation. Figure 17, Figure 18 and Figure 19 show the time domain waveform at the amplitude coefficient $f_{ce}$ of 1.2 N, 8 N, and 100 N. The figures show that the periodic out-of-balance force changes the load distribution, and this changing ability is related to the amplitude of the out-of-balance force; this phenomenon differs from that of the constant load. The peak value of the rolling element–raceway contact force changed under a weak outer race unbalancing effect. In addition to changing periodically, the peak value change frequency also doubled when the effect was significant. The results of other amplitude coefficients at #2 and #7 are illustrated in Figure 20 and Figure 21; the second peak can be observed in the red dotted frame in the figures. We can see that the force amplitude increases as $f_{ce}$ increases.

The obtained contact forces were used to calculate the angular position when F>0. The time domain chart of the load region is shown in Figure 22, Figure 23, Figure 24 and Figure 25, and the areal map of the load region is shown in Figure 26. Figure 22 shows that the load region was constant in the heavy load case. When the unbalance force increased, the load region started to fluctuate. The figures show that the load region periodically changed with time while its extent varied with the out-of-balance force. The load region can produce a slight fluctuation because of the weak unbalance force, and the extent of the load region is increased from 0° to 360° when the unbalance force is great.

For the light load case, we introduced the load region variation through the outer race unbalancing effect. The above results indicate that the unbalance effect can change the load distribution of the bearing; thus, the load region becomes time-varying. Moreover, the larger the unbalance effect is, the more the load region changes. The load region changes in two ways: time domain fluctuation and extent amplification; these changes are the probable causes of the modulation sideband phenomenon described in the previous section.

#### 3.3.3. Analysis of Modulation Sidebands of Cage-Ball Friction Fault under Different Unbalance Effects

We examined the outer race unbalancing effect on the vibration responses by the vertical acceleration. The vibration responses of the cage-ball friction fault under different out-of-balance forces were simulated and analyzed. The amplitude coefficient $f_{ce}$ of the out-of-balance force ranged from 1 N to 100 N. The simulation results when $f_{ce}$ was 1.2 N, 8 N, and 60 N are shown in Figure 27, Figure 28, Figure 29, Figure 30, Figure 31 and Figure 32. The results suggest that the unbalance effect considerably affected the modulation sidebands of the cage–ball friction fault. As the unbalance increased, the second-order sidebands of the cage frequency gradually became more dominant as the load region variation became more obvious. As shown in Figure 32, the two modulation frequencies marked by the red dotted lines are very obvious.

For the 1X harmonic component of the ball spin frequency, we calculated the ratios of the second-order sidebands to the first-order sidebands of the cage frequency under different unbalancing effects. The ratios of the lower sideband are illustrated in Figure 33, while those of the upper sideband are illustrated in Figure 34. The horizontal coordinate axis represents the amplitude coefficient for the out-of-balance force, the red line represents the envelope spectral ratio value, and the blue lines represent the peak value at the two sidebands. The second-order sideband curves rose more distinctly than those of the first order. Although the amplitude relation between the spectral ratio and amplitude coefficient does not appear linear, we can still conclude that both quantities are highly positively correlated, which makes it possible to recognize the severity of the cage friction fault. These specific fault characteristics are crucial for understanding and diagnosing the cage–ball friction fault. As shown by the dotted arrows in Figure 33 and Figure 34, the ratio of the second order side-band to the first order sideband increases with the increase of the unbalanced force.

## 4. Experimental Verifications

To verify the validity of the cage–ball friction fault simulation, we performed micro-vibration test monitoring for the satellite flywheel bearing subassembly. A bearing sub-assembly used for the test had a cage with some slight friction and balls with surface damage but without obvious wear debris in the raceway. Another bearing subassembly had severe cage–ball friction; Figure 35(a1) shows the internal friction condition, which exhibits three distinct characteristics. Figure 35(a2) shows the test rig. First, a large amount of pasted gray-black debris adheres to the ball surface and raceway; second, obvious black wear tracks are on the cage pocket and guiding surface; third, the outer and inner raceways have discoloration indentations. The bearing characteristic frequencies of the test bearings are presented in Table 3.

For the monitoring results of the bearings with slight amounts of friction, Figure 36, Figure 37 and Figure 38 show the acceleration time-domain waveforms, spectrum, and envelope spectrum, respectively. Figure 39, Figure 40 and Figure 41 illustrate the corresponding monitoring results of the severe cage-ball friction. As shown in Figure 38, many multiple frequency components exist, such as the cage revolution and ball spin frequencies. In addition, the obvious first-order modulation sidebands of $f_{c}$ can be observed. Likewise, it can be observed that the second-order sidebands of $f_{c}$ are more obvious than those of the first-order sidebands. The variation behavior of the modulation sidebands in the simulation is similar to that in the experiment. Thus, the test results verify the accuracy of the simulation results to a certain extent.

The sideband ratios graph is shown in Figure 42. The spectral ratio was low when the cage friction fault was slight, and the ratio became significantly high when the cage friction fault was severe. This phenomenon qualitatively reflects the highly positive correlation between the ratio and the severity of the cage friction fault.

## 5. Conclusions

This paper presents a variable stiffness method based on a 2-DOF bearing model for the cage–ball friction fault of lightly loaded bearings. Considering the load region variation resulting from the unbalancing effect of the outer race, we examined the dynamic response of the bearings by simulation and performed experiments for validation. According to the investigation results, the following are the major findings of this work:(1)The stiffness change factor has a great impact on the vibration response. If the factor becomes larger, the vibration energy also becomes larger. It clearly explains the transmission of the bearing failure energy. Therefore, the variable stiffness method is effective for simulating friction faults between the cage and rolling elements.(2)When the cage–ball friction fault occurs, many frequency components exist in the envelope spectrum. In addition to the ball spin, cage revolution, and the multiple frequency components, the cage frequency has certain characteristic modulation sidebands. For the heavy load case, the first-order sidebands are highly obvious. For the light load case, the second-order sidebands are more obvious than the first-order sidebands while the cage friction severity grows.(3)The modulation frequency components of the cage and rolling elements change with the severity of the fault. Therefore, a modulation sideband ratio method based on envelope spectrum is proposed for qualitatively diagnosing the severity of cage-rolling element friction faults, and the effectiveness of the presented method is verified by the experiments. No matter the simulation or the experiment, the spectral ratio was low when the cage friction fault was slight, and the ratio became significantly higher when the cage friction fault was severe.

In the future, we will try to establish a simulation model considering the detailed friction dynamic process to quantitatively evaluate the characteristics under different friction conditions, which will be used to quantitatively evaluate the degree of friction faults.

## Figures and Tables

**Figure 1 sensors-22-08346-f001:**
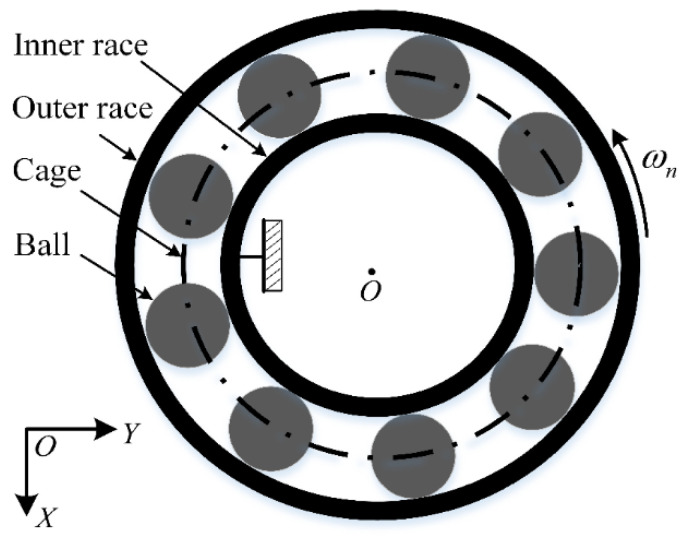
Rolling element bearing system.

**Figure 2 sensors-22-08346-f002:**
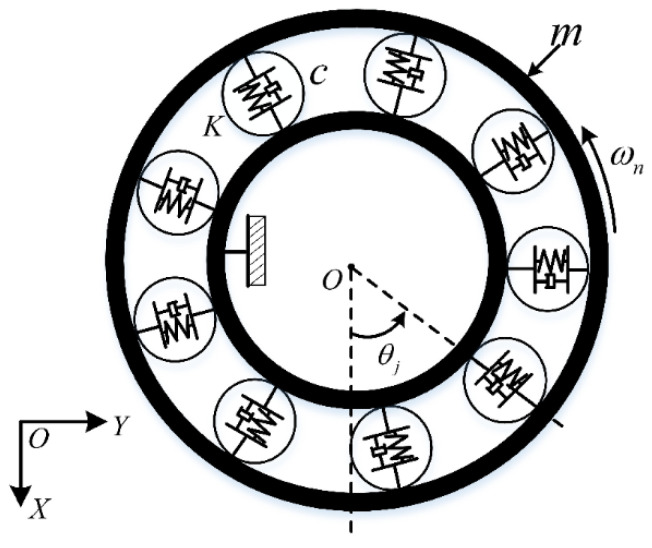
Spring–damp dynamic model of rolling element bearing.

**Figure 3 sensors-22-08346-f003:**
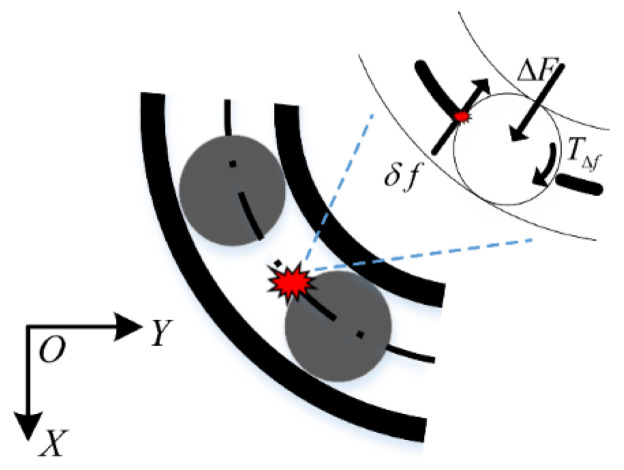
Schematic of friction fault.

**Figure 4 sensors-22-08346-f004:**
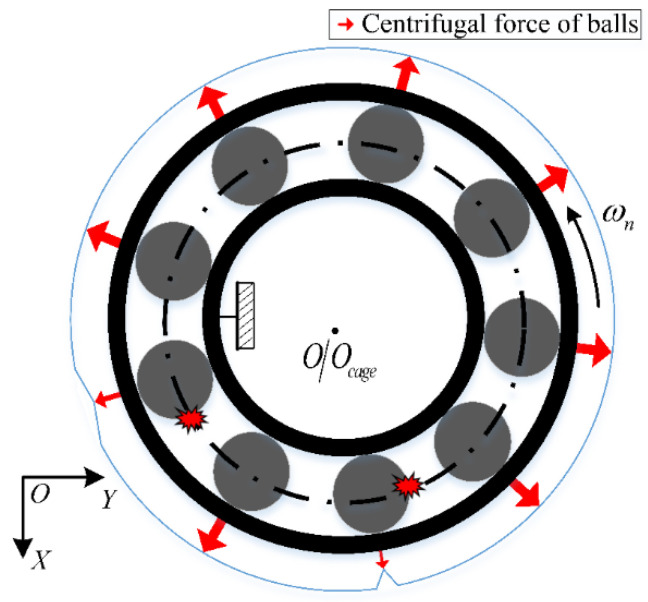
Schematic of centrifugal force of balls with friction fault.

**Figure 5 sensors-22-08346-f005:**
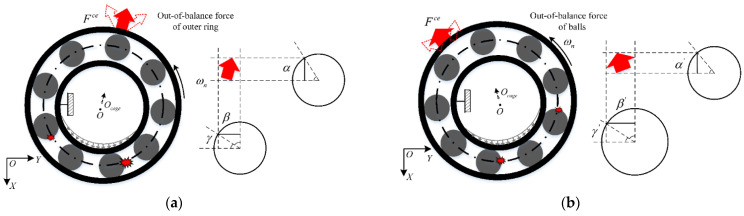
Schematic of out-of-balance force of outer ring: (**a**) time elapsed t; (**b**) time elapsed t+Δt.

**Figure 6 sensors-22-08346-f006:**
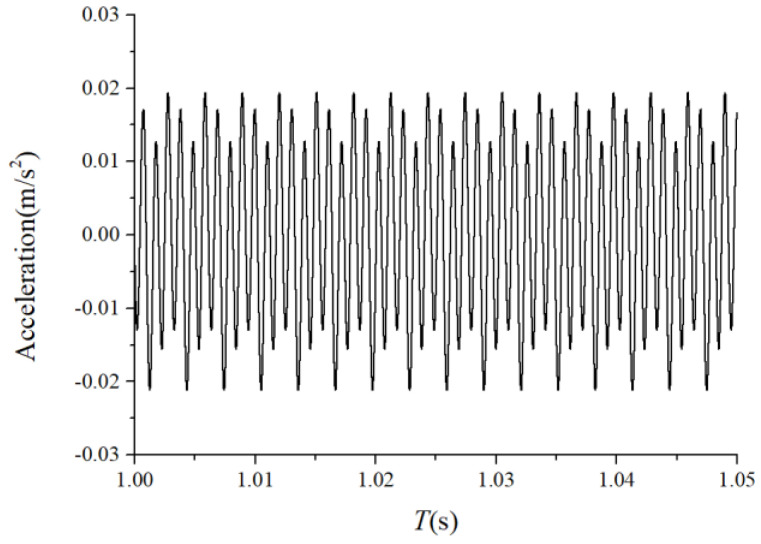
Acceleration waveform (normal condition).

**Figure 7 sensors-22-08346-f007:**
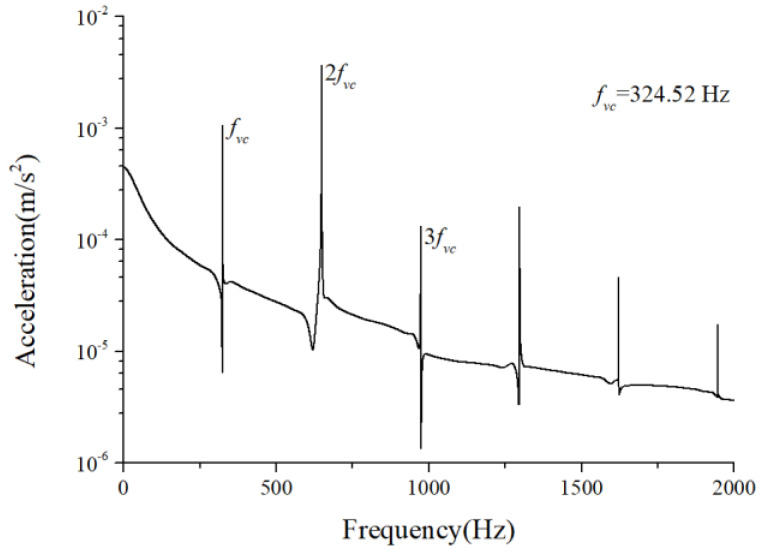
Acceleration envelope spectrum (normal condition).

**Figure 8 sensors-22-08346-f008:**
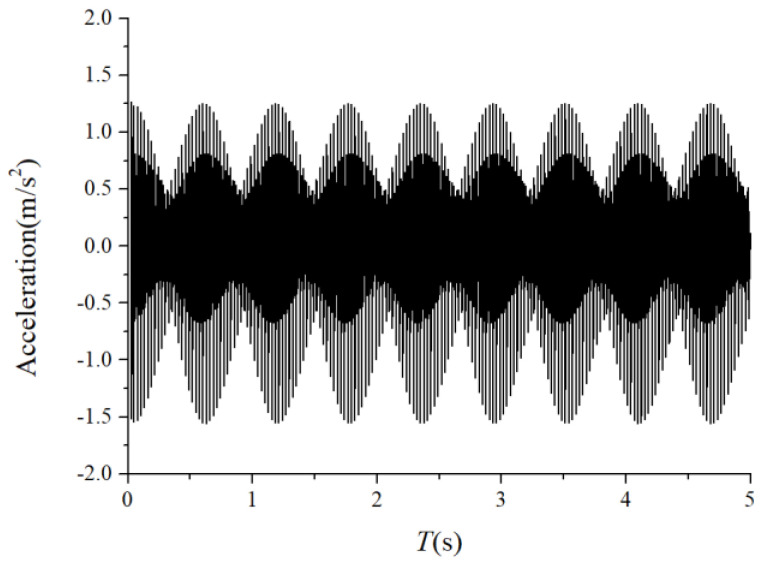
Acceleration waveform.

**Figure 9 sensors-22-08346-f009:**
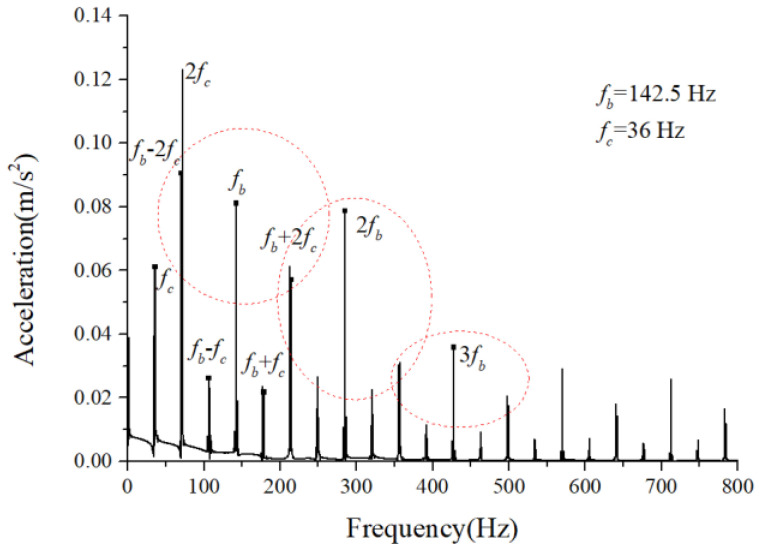
Acceleration envelope spectrum.

**Figure 10 sensors-22-08346-f010:**
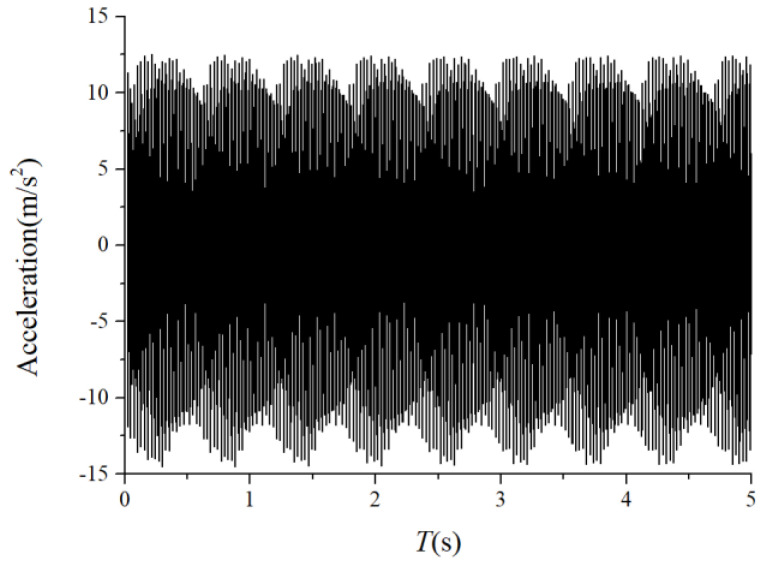
Acceleration waveform.

**Figure 11 sensors-22-08346-f011:**
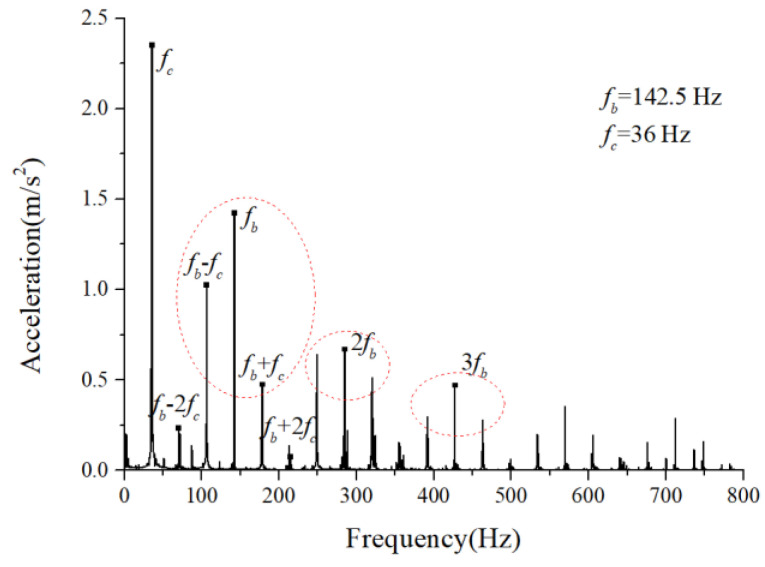
Acceleration envelope spectrum.

**Figure 12 sensors-22-08346-f012:**
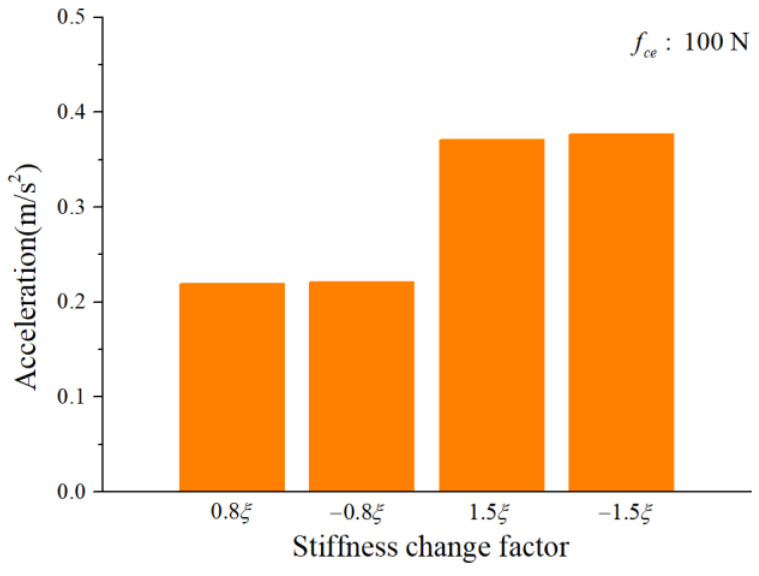
Acceleration RMS.

**Figure 13 sensors-22-08346-f013:**
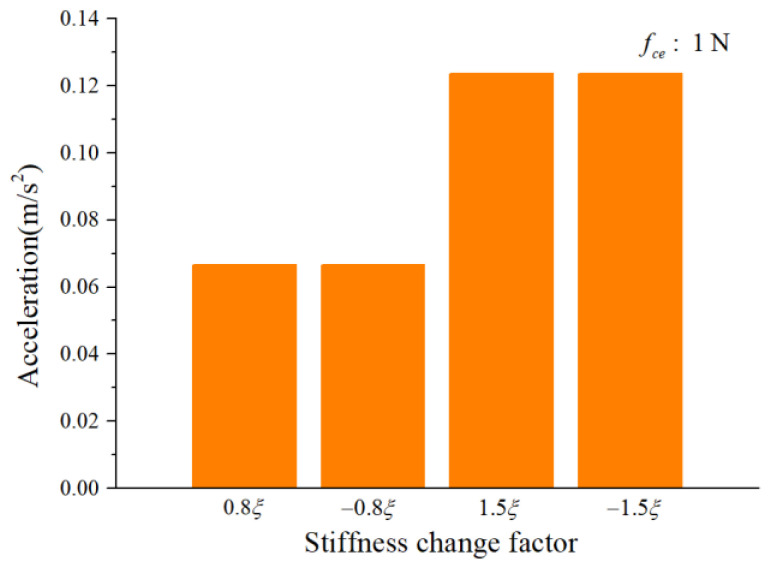
Acceleration RMS.

**Figure 14 sensors-22-08346-f014:**
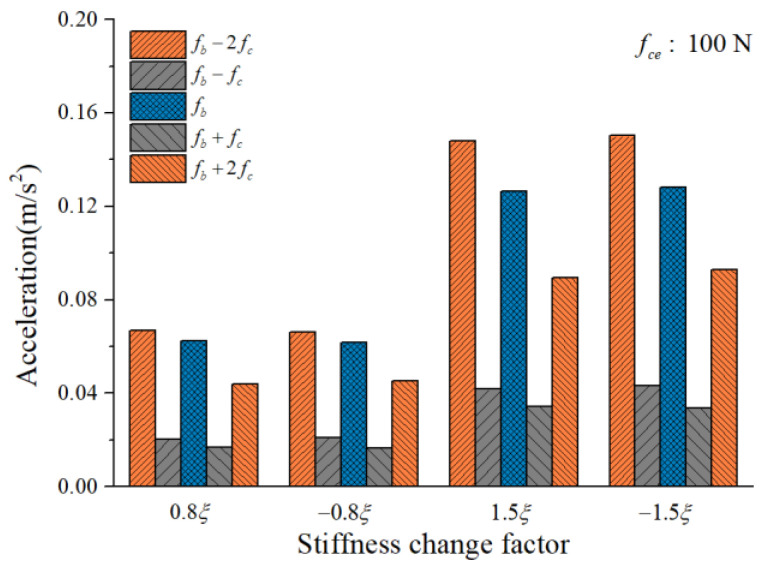
Envelope spectrum peak value at sideband modulation family.

**Figure 15 sensors-22-08346-f015:**
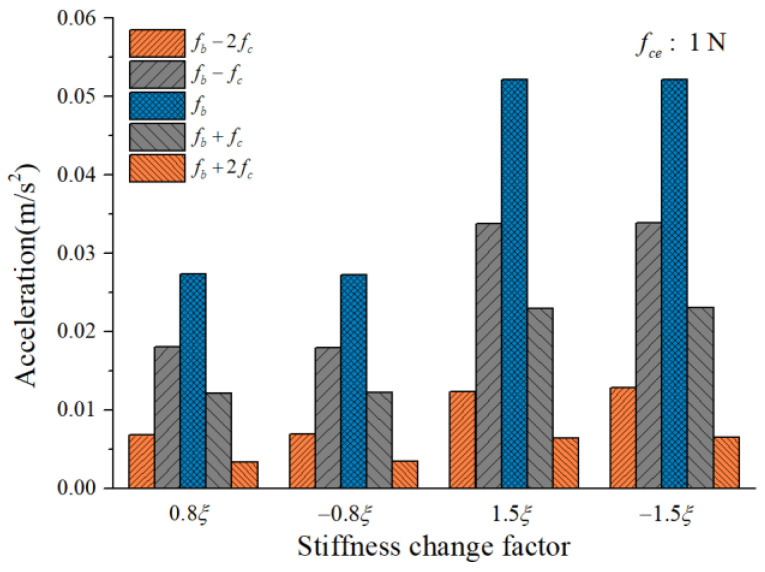
Envelope spectrum peak value at sideband modulation family.

**Figure 16 sensors-22-08346-f016:**
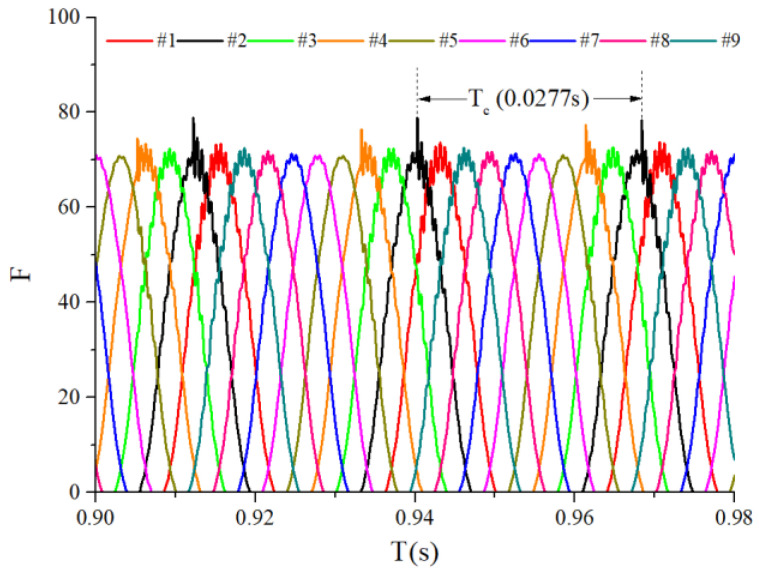
Time-domain waveform of the heavy load case contact force.

**Figure 17 sensors-22-08346-f017:**
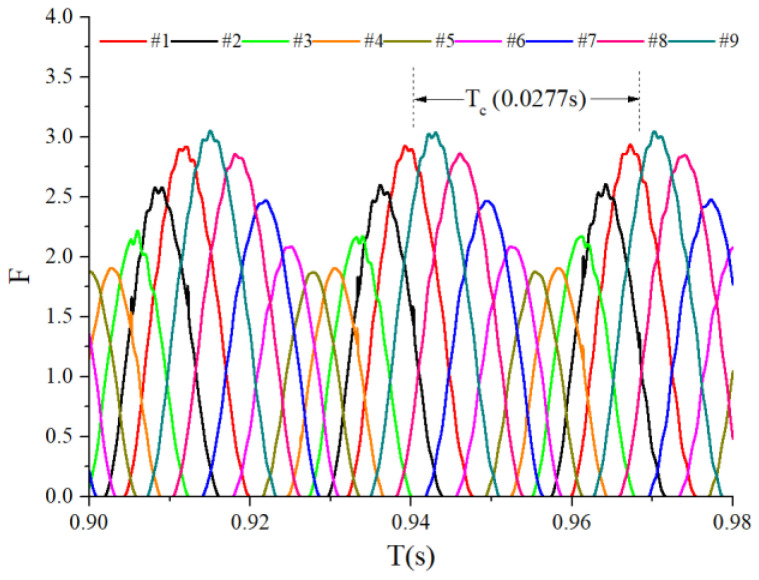
Time-domain waveform of the contact force ($f_{ce}$ = 1.2 N).

**Figure 18 sensors-22-08346-f018:**
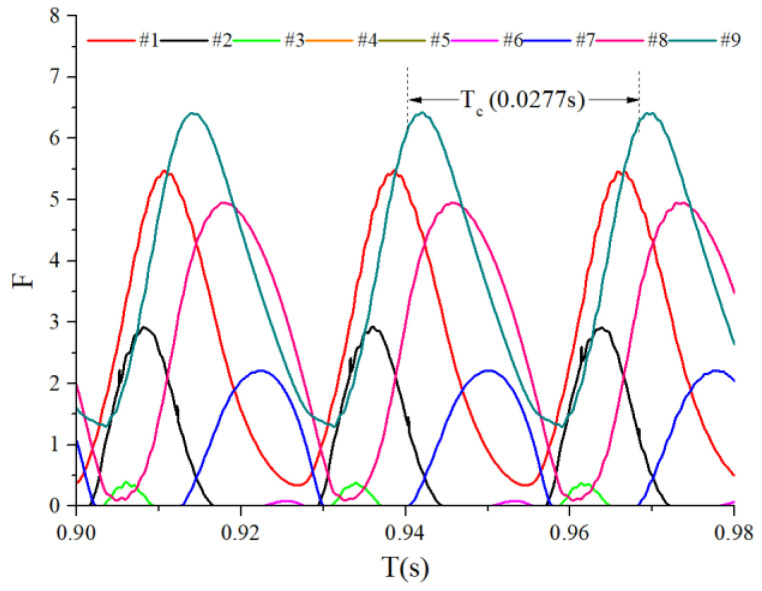
Time-domain waveform of the contact force ($f_{ce}$ = 8 N).

**Figure 19 sensors-22-08346-f019:**
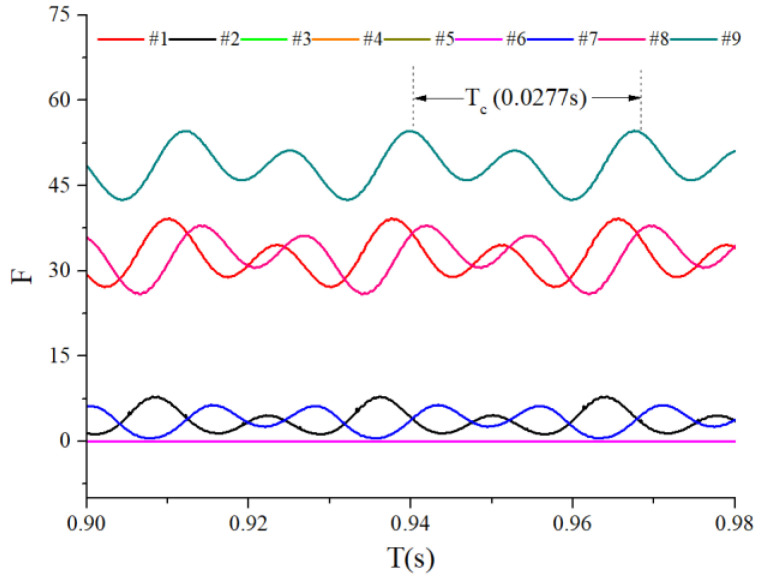
Time-domain waveform of the contact force ($f_{ce}$ = 100 N).

**Figure 20 sensors-22-08346-f020:**
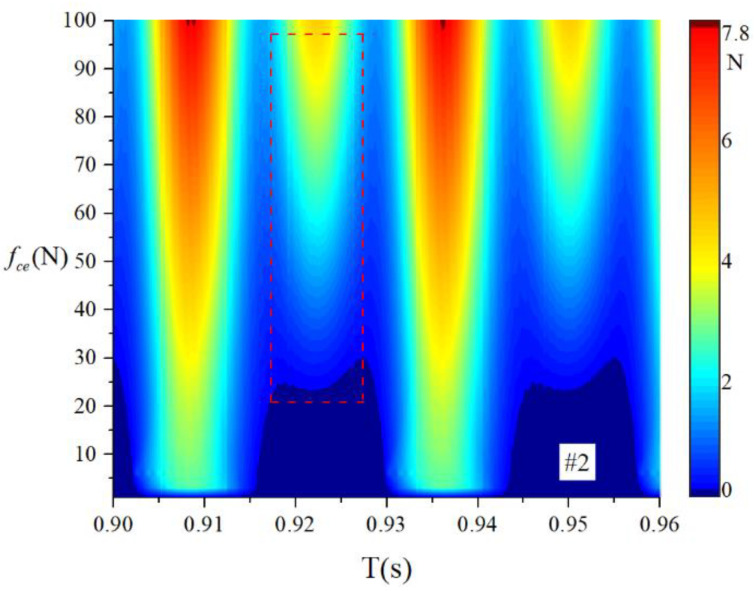
Waterfall of the rolling element–raceway contact force (#2).

**Figure 21 sensors-22-08346-f021:**
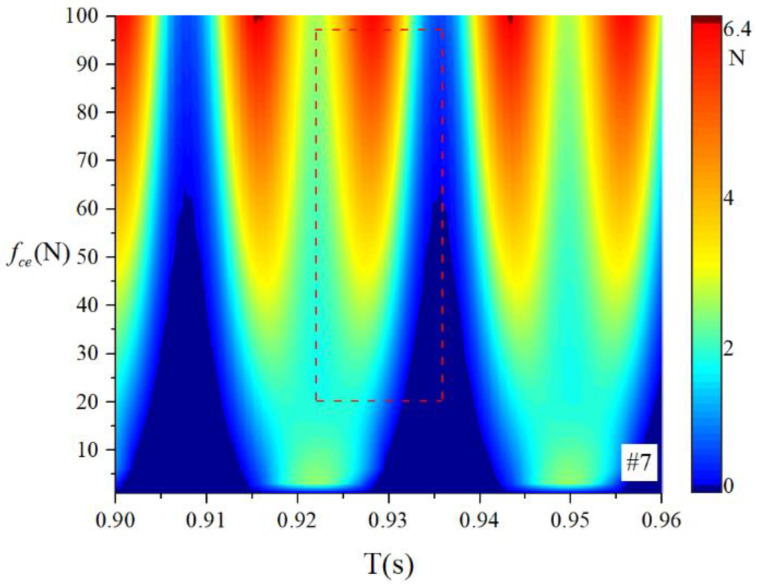
Waterfall of the rolling element–raceway contact force (#7).

**Figure 22 sensors-22-08346-f022:**
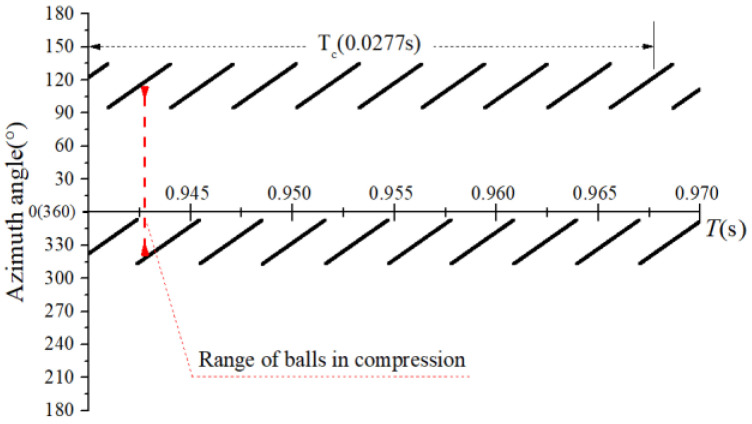
Time-domain diagram of the heavy load case load region.

**Figure 23 sensors-22-08346-f023:**
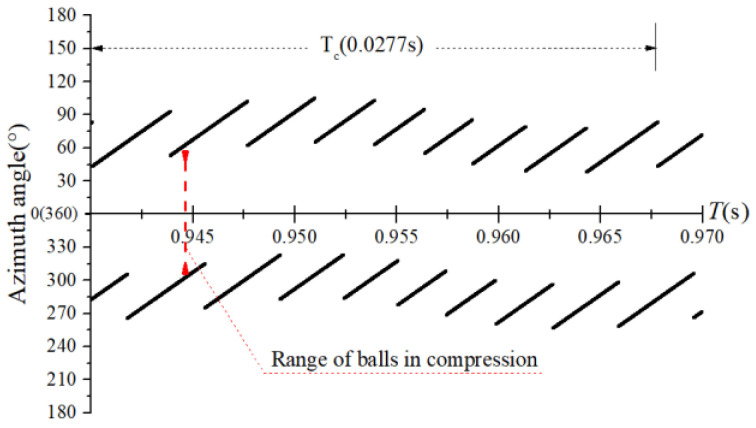
Time-domain diagram of the load region ($f_{ce}$ = 1.2 N).

**Figure 24 sensors-22-08346-f024:**
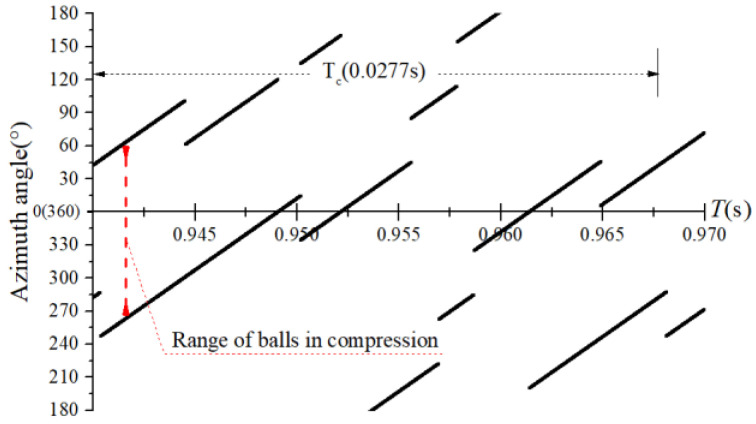
Time-domain diagram of the load region ($f_{ce}$ = 8 N).

**Figure 25 sensors-22-08346-f025:**
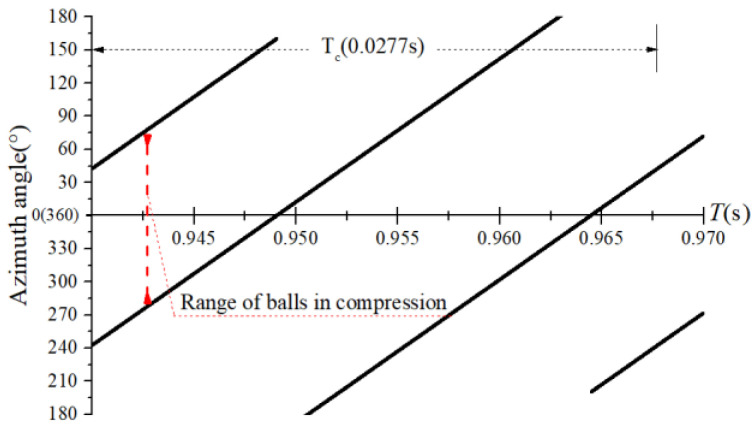
Time-domain diagram of the load region ($f_{ce}$ = 100 N).

**Figure 26 sensors-22-08346-f026:**
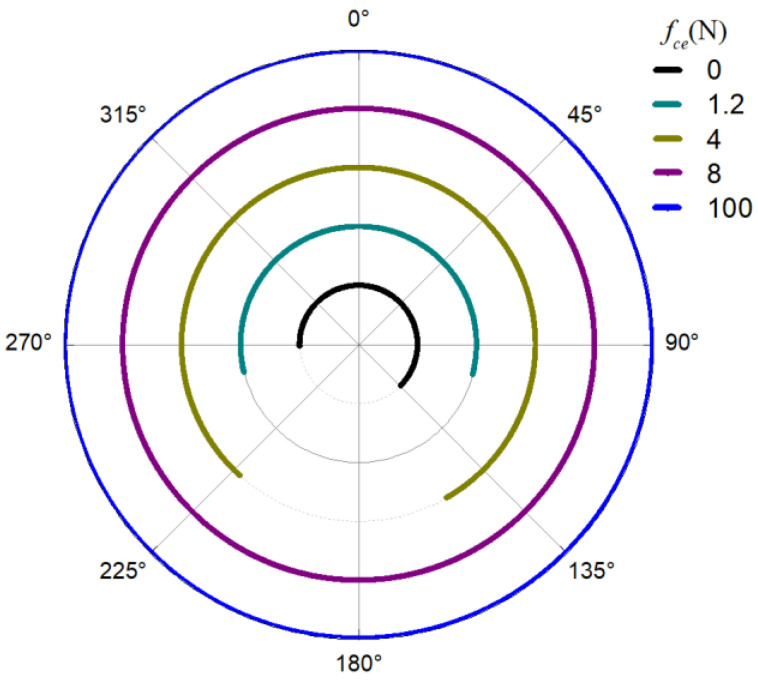
Areal map of the load region.

**Figure 27 sensors-22-08346-f027:**
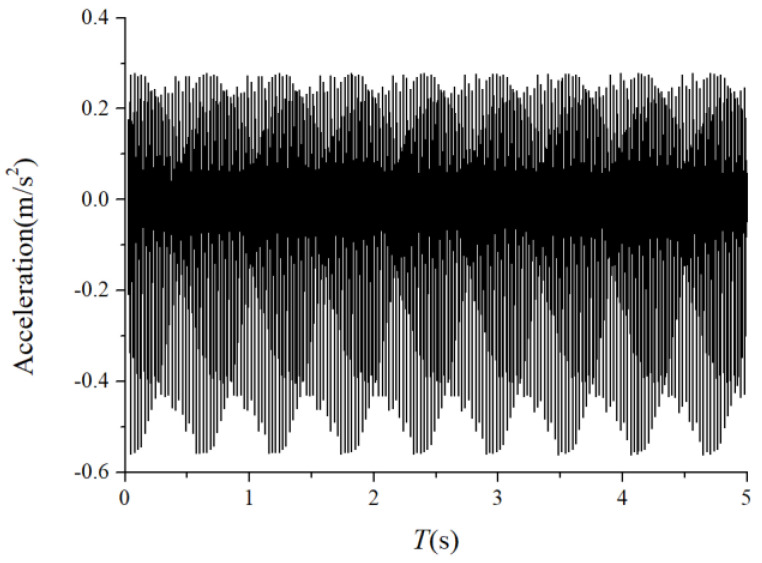
Acceleration waveform.

**Figure 28 sensors-22-08346-f028:**
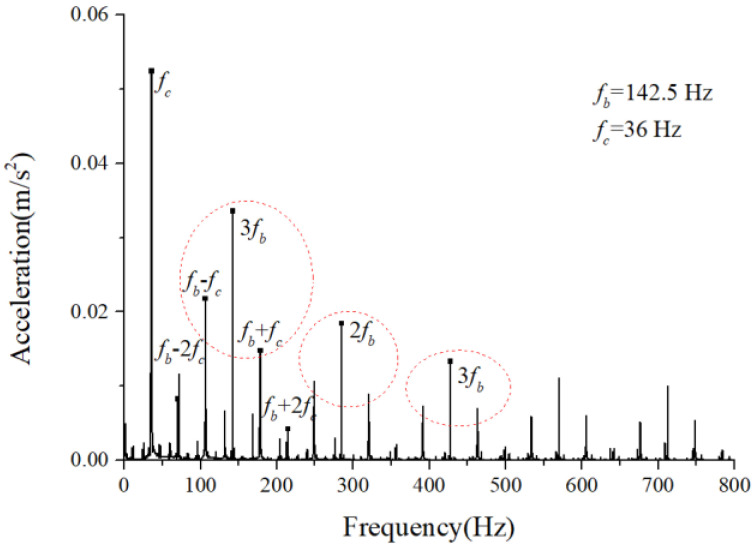
Acceleration envelope spectrum.

**Figure 29 sensors-22-08346-f029:**
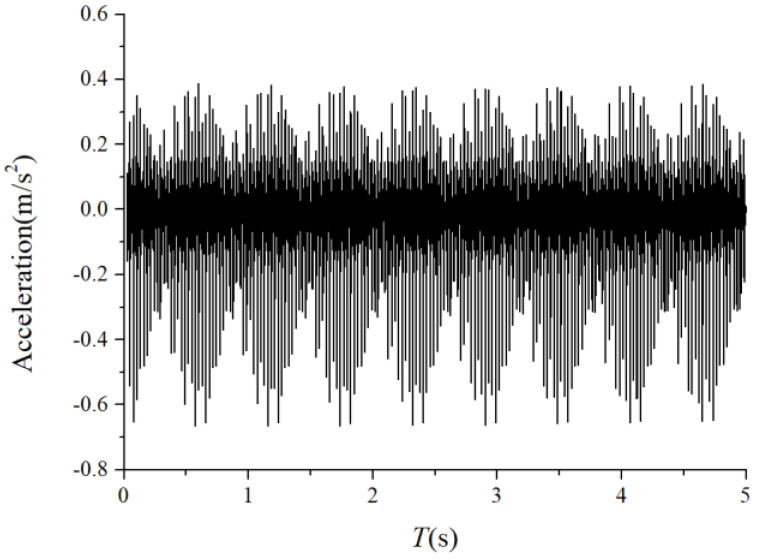
Acceleration waveform.

**Figure 30 sensors-22-08346-f030:**
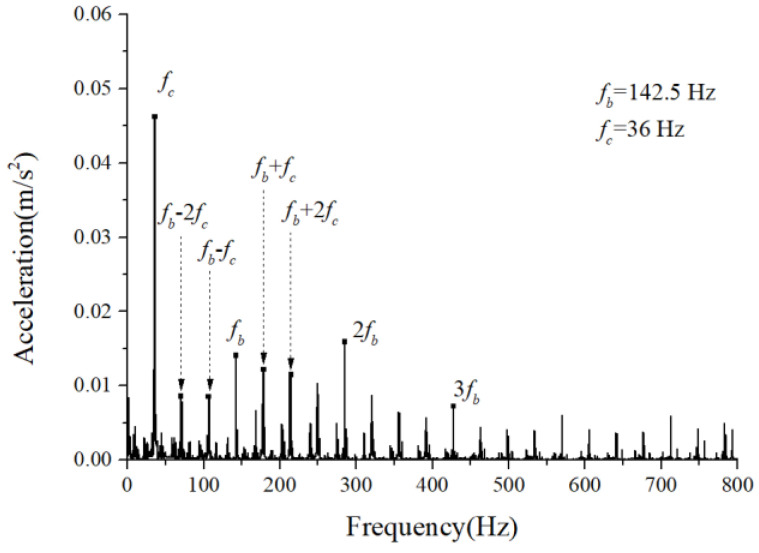
Acceleration envelope spectrum.

**Figure 31 sensors-22-08346-f031:**
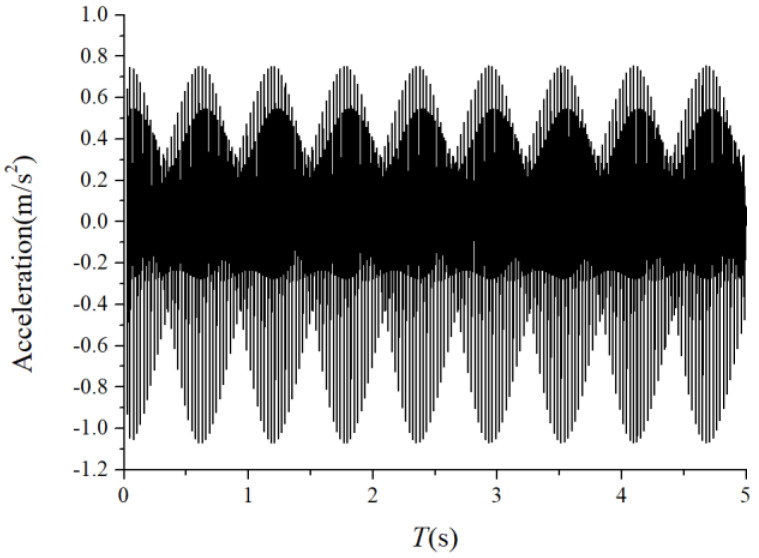
Acceleration waveform.

**Figure 32 sensors-22-08346-f032:**
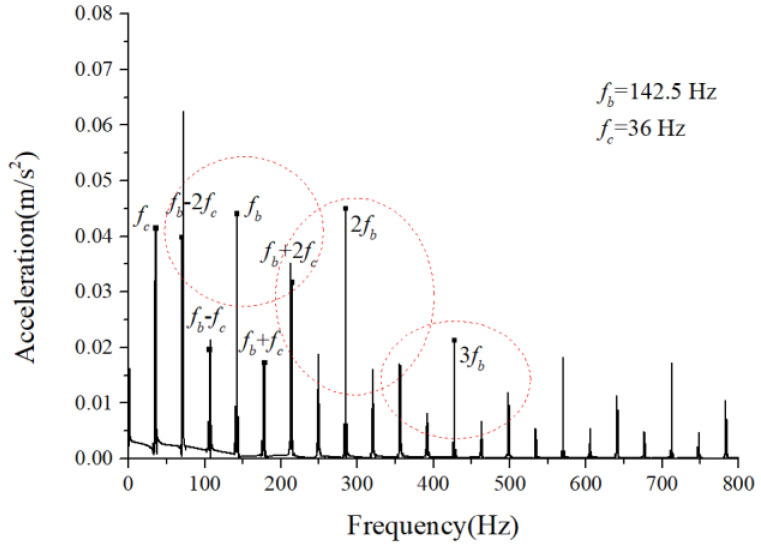
Acceleration waveform.

**Figure 33 sensors-22-08346-f033:**
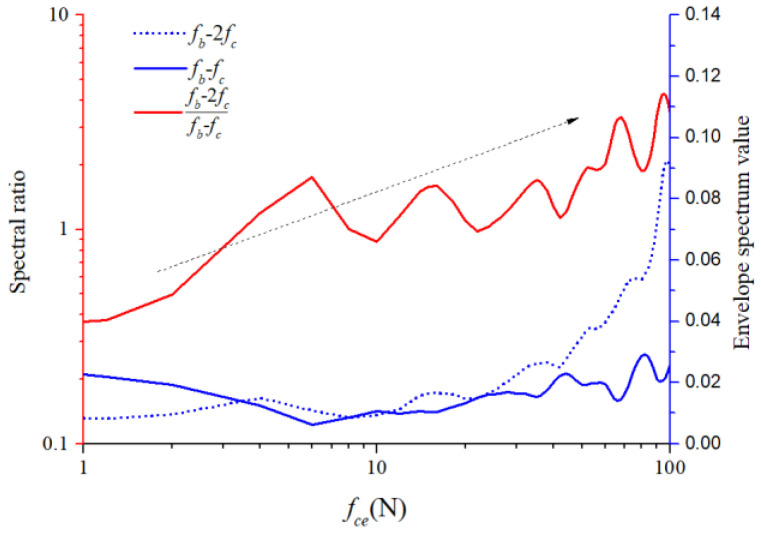
Modulation of lower sideband graph.

**Figure 34 sensors-22-08346-f034:**
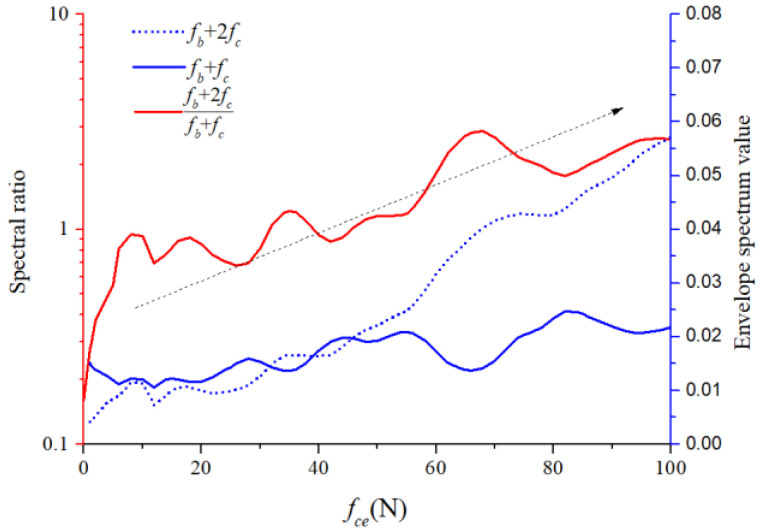
Modulation of upper sideband graph.

**Figure 35 sensors-22-08346-f035:**
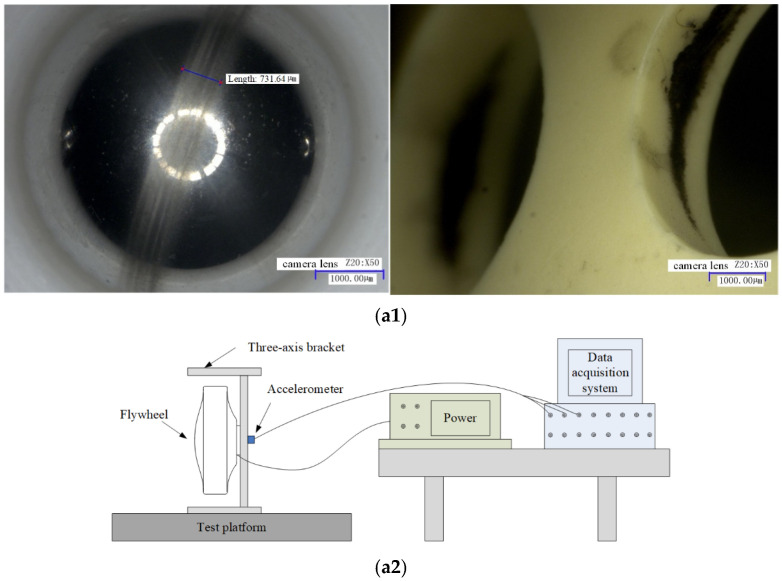
(**a1**) Disassembly diagram. (**a2**) Diagram of test rig.

**Figure 36 sensors-22-08346-f036:**
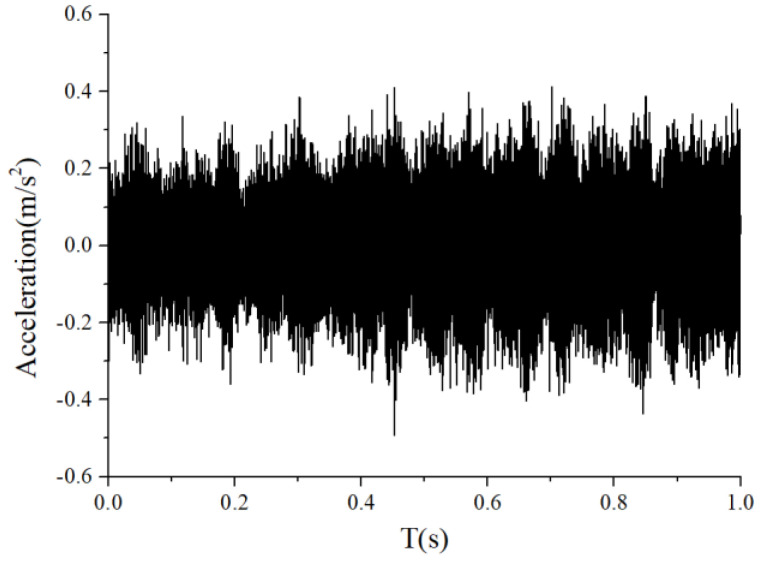
The time-domain waveform of the test signal (slight friction).

**Figure 37 sensors-22-08346-f037:**
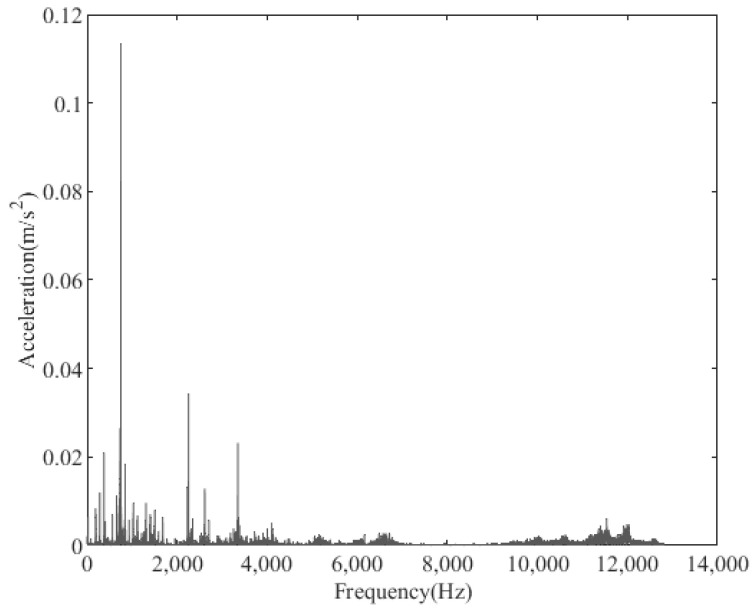
The amplitude spectrum of the test signal (slight friction).

**Figure 38 sensors-22-08346-f038:**
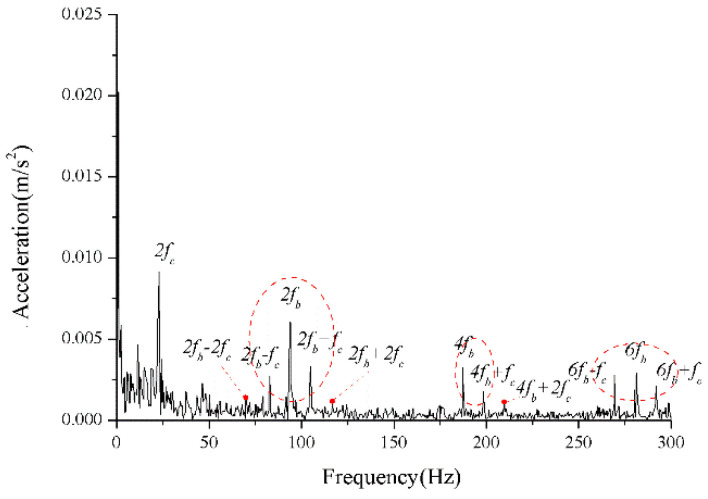
The envelope spectrum of the test signal (slight friction).

**Figure 39 sensors-22-08346-f039:**
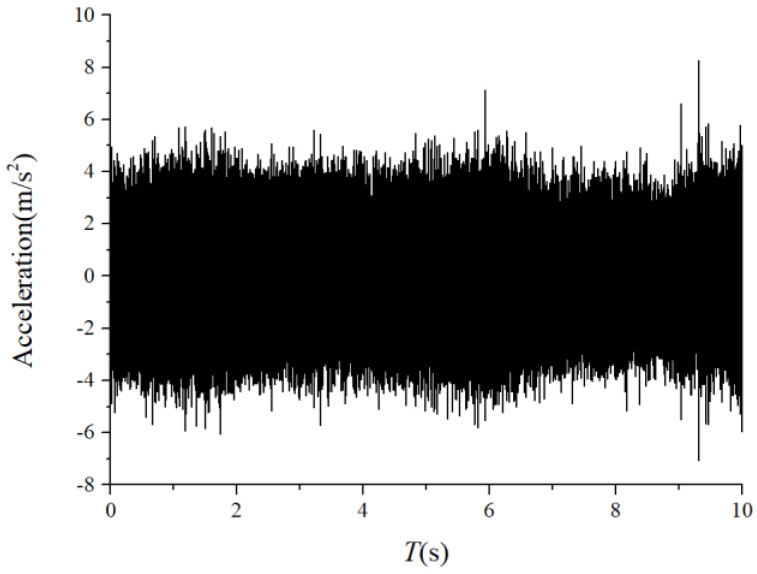
The time-domain waveform of the test signal (severe friction).

**Figure 40 sensors-22-08346-f040:**
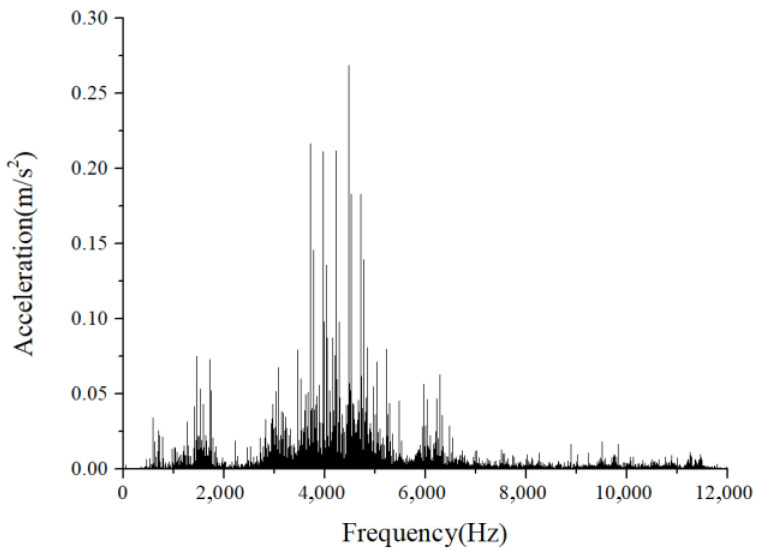
The amplitude spectrum of the test signal (severe friction).

**Figure 41 sensors-22-08346-f041:**
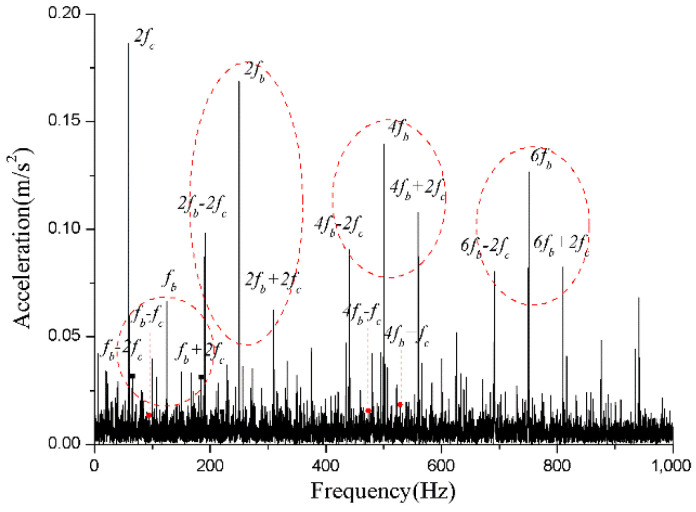
The envelope spectrum of the test signal (severe friction).

**Figure 42 sensors-22-08346-f042:**
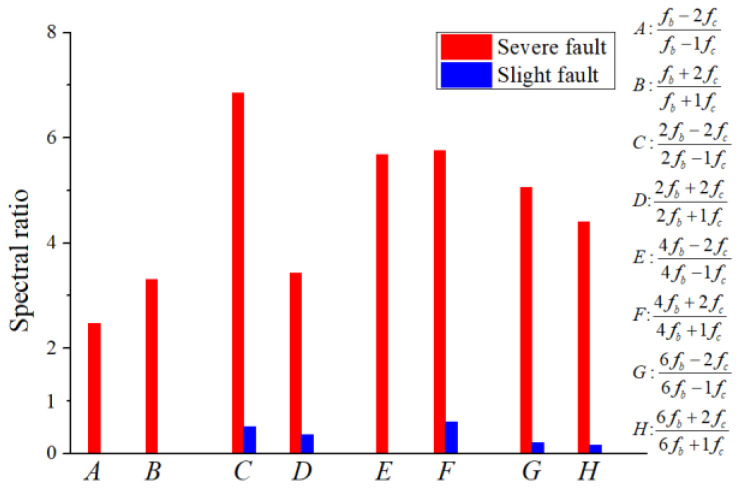
Modulation sidebands graph (test).

**Table 1 sensors-22-08346-t001:** Deep groove bearing parameters of MB-ER-16K [59].

Parameter	Value
Ball stiffness (K)	$9.39\times 10^{9}{\;N/m}^{1.5}$
Pitch diameter (D)	39.32 mm
$\mathrm{Ball}\;\mathrm{diameter}\;(D_{b}$)	7.94 mm
Number of balls (N)	9
Outer race mass (m)	0.5 kg
Outer race damping (c)	200 Ns/m
Time of unit impulse (τ)	0.1 ms
Radial clearance (ε)	0 mm (assumed)
Stiffness change factor (ξ)	0.1
Phase difference (φ)	$5^{\circ }$
$\mathrm{Coefficient}\;(c_{1}$)	0
$\mathrm{Coefficient}\;(c_{2}$)	100

**Table 2 sensors-22-08346-t002:** Deep groove bearing frequencies of MB-ER-16K.

Parameter	Value
$\mathrm{Ball}\;\mathrm{spin}\;\mathrm{frequency}\;(f_{b}$)	$2.375f_{n}$
$\mathrm{Cage}\;\mathrm{revolution}\;\mathrm{frequency}\;(f_{c}$)	$0.601f_{n}$
$\mathrm{VC}\;\mathrm{frequency}\;(f_{VC}$)	$5.409f_{n}$

**Table 3 sensors-22-08346-t003:** Bearing characteristic frequencies of test bearings.

Parameter	Value
$\mathrm{Ball}\;\mathrm{spin}\;\mathrm{frequency}\;(f_{b}$)	$2.494f_{n}$
$\mathrm{Cage}\;\mathrm{revolution}\;\mathrm{frequency}\;(f_{c}$)	$0.594f_{n}$

## Data Availability

Not applicable.

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
