# Peer review of "Simulation of Friction Fault of Lightly Loaded Flywheel Bearing Cage and Its Fault Characteristics"

_sensors, 2022, doi:10.3390/s22218346_

Round 1

Reviewer 1 Report

The work proposes a modulation sideband ratio method based on envelope spectrum for qualitatively diagnosing the severity of cage-rolling element friction faults The proposed approach is ok and some important issues should be explained:

1.     Many references are a bit out of date. Current issues and challenges should be analyzed in INTRODUCTION.

2.     The content of each chapter should be introduced in INTRODUCTION.

3.     The proposed model uses various assumptions many times. What are these assumptions based on? How do these assumptions affect the simulation results?

4.     Recent fault diagnosis related work could be referred to, Data-driven early fault diagnostic methodology of permanent magnet synchronous motor, Optimal sensor placement methodology of hydraulic control system for fault diagnosis, Fault detection and diagnostic method of diesel engine by combining rule-based algorithm and BNs/BPNNs, Remaining useful life re-prediction methodology based on Wiener process: Subsea Christmas tree system as a case study.

5.     There are many interesting figures in the Section 3. But these figures are not sufficiently explained.

6.     How do the simulations performed contribute to bearing fault diagnosis?

Author Response

Dear reviewer:

Great thanks for your good comments and suggestions. We’ve made necessary modifications as follows:

  1. Many references are a bit out of date. Current issues and challenges should be analyzed in INTRODUCTION.

Response:

We have added recent research literatures, and descripted them as follows:

“The extraction of significant features is essential for efficient fault diagnosis and prognosis of rolling element bearing [4]. Zhang et al. [5] proposed a periodic low rank dynamic mode decomposition algorithm to extract features. Cui et al. [6] used the machine learning to extract the bearing fault features. Wang et al. [7] proposed a robust fault characteristic extraction approach based on the time-frequency analysis. Cai et al. [8] proposed a rule-based algorithm and Bayesian networks (BNs) or Back Propagation neural networks. Cai et al. [9] contributed a RUL re-prediction method based on Wiener process. Kong et al. [10] proposed a sensor placement methodology of hydraulic control system. Cai et al. [11] proposed a rule-based algorithm and Bayesian networks (BNs) or Back Propagation neural networks. These studies show that capturing the specific fault characteristics of bearings is conducive to better diagnosis, and feature extraction is one of the current research focuses.”

  1. The content of each chapter should be introduced in INTRODUCTION.

Response:

Thank you for your good suggestions.

We have added the content of each chapter at the end of INTRODUCTION:

“The remainder of the paper is organized as follows: In the Section 2, the 2D model of cage-ball friction fault for lightly loaded bearings is established; In the Section 3, based on the proposed simulation model, the features of the cage friction fault were analyzed under different key parameters; In the Section 4, the features of the cage friction fault were confirmed by experiment. Finally, the conclusions of this paper is summarized in the Section 5.”

  1. The proposed model uses various assumptions many times. What are these assumptions based on? How do these assumptions affect the simulation results?

Response:

The purpose of this paper is to obtain the qualitative characteristics of cage friction fault and provide guidance for fault diagnosis. In order to facilitate calculation without losing the nature of fault characteristics, the following assumptions are made in this paper:

Hypothesis 1: Hertz contact deformation theory hypothesis, which affects the calculation method of bearing force.

Hypothesis 2: The rigid ball hypothesis assumes that the rolling element is rigid.

Assumption 3: When the cage and rolling element have friction failure, the friction can be transferred to the rolling element and raceway.

It is worth mentioning that we use the change of stiffness to simulate the change of friction.

These assumptions have no substantial impact on the qualitative analysis of fault characteristics. Of course, these assumptions need to be revised if quantitative analysis of fault characteristics is required.

  1. Recent fault diagnosis related work could be referred to, Data-driven early fault diagnostic methodology of permanent magnet synchronous motor, Optimal sensor placement methodology of hydraulic control system for fault diagnosis, Fault detection and diagnostic method of diesel engine by combining rule-based algorithm and BNs/BPNNs, Remaining useful life re-prediction methodology based on Wiener process: Subsea Christmas tree system as a case study.

Response:

Thank you for your suggestion. We have added these articles in INTRODUCTION as [8, 10, 11].

  1. There are many interesting figures in the Section 3. But these figures are not sufficiently explained.

Response:

Thank you for your suggestion. Now, all figures are vector images with 300 dpi resolution.

  1. How do the simulations performed contribute to bearing fault diagnosis?

Response:

According to the traditional fault diagnosis method, the cage has its own fault characteristic frequency. But in the light load bearing friction fault, it is difficult to find the corresponding characteristic frequency in the frequency spectrum, or even find the characteristic frequency of the local defect of the ball. This brings troubles to the diagnosis of cage friction faults.

Based on the research of this paper, the characteristic frequencies of cage friction fault are discovered. When a friction fault occurs between the cage and the rolling element, the frequency of the rotating frequency component of the cage will modulate the rotating frequency component of the rolling element, that is, the side frequency components on both sides of the rolling element characteristic frequency (separated by the cage characteristic frequency). The modulation frequency component of cage and rolling element will change with the severity of the fault. Therefore, it can provide guidance for diagnosis.

The authors would like to thank you for your valuable comments, which are extremely important for the improvement of this manuscript.

Thanks again.

Kind regards,

The authors

Reviewer 2 Report

I found your article very interesting, but in my opinion below remarks would improve your manuscript under the scientific level.

Comments and Suggestions for Authors:

1.       In the Abstract I miss the most important information about the novelty and outcomes of conducted research. Please don’t focus on the general description of conducted research.

2.       In the Introduction, there is some change of font size.

3.       The literature review seems to be quite old, so I suggested to refer to the papers discussing the modelling problem:

·       Ambrożkiewicz et al. (2021), Analysis of dynamic response of a two degrees of freedom (2-DOF) ball bearing nonlinear model. Applied Sciences, 171, 11(2),
pp. 1-23, 787.

·       Gao et al. (2022), Experimental and theoretical approaches for determining cage motion dynamic characteristics of angular contact ball bearings considering whirling and overall skidding behaviors. Mechanical Systems and Signal Processing, 168, 108704.

4.       In the end of Introduction, I suggest to add the remainder of the paper.

5.       In the Section 2, please introduce the sentence about the rotation of the outer ring in case of flywheel. It can be misunderstood by readers in the future. This is important to describe the operation of bearing in case of flywheel.

6.       Please the refer power 3/2 in set of equations 3, to the point contact between balls and raceways.

7.       Referring to Section 2.2.1, how do you refer the stiffness variations to the real value of friction torque as occur in the rolling-element bearing?

8.       I suggest to substitute word “imbalance” with unbalance. How it is implemented into the equations of motion?

9.       What kind of standardized bearing do you study, the catalogue code is not enough.

10.   The value specified in the mathematical model are not well taken. How do you presume such small damping, as the clearance is 0. It makes no sense. It would make sense if there is some load assumed, which I can’t find.

11.   The experimental verification seems to be conducted well, but please order a little bit the part referring to the results of mathematical model.

12.   In Conclusions you mentioned that tests were conducted under small load, but I can’t find the specified value for it.

13.   Referring to the experimental part, please describe the test rig as it can be very interesting point to the readers.

14.   In the paper I miss the strict comparison between the model results and experiment.

15.   In Conclusions, please specify how do you want to extend your research.

Author Response

Dear editor and reviewers:

Great thanks for your good comments and suggestions. According to helpful comments and suggestions from editors and reviewers, we’ve made necessary modifications as follows:

  1. In the Abstract I miss the most important information about the novelty and outcomes of conducted research. Please don’t focus on the general description of conducted research.

Response:

Thank you for your suggestion. We have revised the abstract.

  1. In the Introduction, there is some change of font size.

Response:

We have revised it.

  1. The literature review seems to be quite old, so I suggested to refer to the papers discussing the modelling problem: Ambrożkiewicz et al. (2021), Analysis of dynamic response of a two degrees of freedom (2-DOF) ball bearing nonlinear model. Applied Sciences, 171, 11(2),pp. 1-23, 787.  Gao et al. (2022), Experimental and theoretical approaches for determining cage motion dynamic characteristics of angular contact ball bearings considering whirling and overall skidding behaviors. Mechanical Systems and Signal Processing, 168, 108704..

Response:

We have added more than 10 articles including these two papers.

  1. In the end of Introduction, I suggest to add the remainder of the paper.

Response:

Thank you for your good suggestions.

We have added the content of each chapter at the end of INTRODUCTION:

“The remainder of the paper is organized as follows: In the Section 2, the 2D model of cage-ball friction fault for lightly loaded bearings is established; In the Section 3, based on the proposed simulation model, the features of the cage friction fault were analyzed under different key parameters; In the Section 4, the features of the cage friction fault were confirmed by experiment. Finally, the conclusions of this paper is summarized in the Section 5.”

  1. In the Section 2, please introduce the sentence about the rotation of the outer ring in case of flywheel. It can be misunderstood by readers in the future. This is important to describe the operation of bearing in case of flywheel.

Response:

We have added the description as:

“The flywheel bearing is different from ground bearings, namely, the operation of flywheel bearing is the outer race.”

  1. Please the refer power 3/2 in set of equations 3, to the point contact between balls and raceways.

Response:

In the literature [59], the refer power 3/2 is for ball bearing, and the refer power 10/9 is for roller bearing. In this paper, we study the ball bearing, so the refer power should be 3/2.

The classical Hertz contact formula is used here. It may be different from the actual situation, but does not affect the qualitative analysis in this paper, especially the characteristic frequency. If we want to obtain the exact amplitude of the characteristic frequencies, we need to further determine the accuracy of the power.

[59] C. Mishra, A.K. Samantaray, G. Chakraborty, Ball bearing defect models: A study of simulated and experimental fault signatures, Journal of Sound and Vibration, Volume 400, 2017, Pages 86-112.

  1. Referring to Section 2.2.1, how do you refer the stiffness variations to the real value of friction torque as occur in the rolling-element bearing?

Response:

When the cage friction fault occurs,  force will be generated, and further transferred to  reaction force between roller and raceway. Hertz contact stiffness  describes the contact force between the rolling element and the raceway. Therefore,  can be expressed by the change of stiffness . Although their amplitudes cannot be measured, we can study the qualitative relationship between them.

  1. I suggest to substitute word “imbalance” with unbalance. How it is implemented into the equations of motion?.

Response:

The “imbalance” was replaced by “unbalance”.

  1. What kind of standardized bearing do you study, the catalogue code is not enough.

Response:

The deep groove bearing typed MB-ER-16K is used for the simulation, and its parameters in literature [59] are referenced. All parameters required for simulation are listed in Table 1.

  1. The value specified in the mathematical model are not well taken. How do you presume such small damping, as the clearance is 0. It makes no sense. It would make sense if there is some load assumed, which I can’t find.

Response:

In this paper, the research is based on the steady-state solution of dynamic simulation, such as envelope spectrum Fig 7 is from Fig 6, and Fig 6 can be regarded as a stable periodic response. Therefore, damping is not critical for fault feature extraction.

If the amplitudes of characteristic frequencies are to be further quantitatively analyzed, damping is very important. This will be the future research work.

  1. The experimental verification seems to be conducted well, but please order a little bit the part referring to the results of mathematical model.

Response:

When the cage–ball friction fault occurs, many frequency components will exist in the envelope spectrum. In addition to the ball spin, cage revolution, and the multiple frequency components, the cage frequency has certain characteristic modulation sidebands. For the light load case, the second-order sidebands are more obvious than the first-order sidebands while the cage friction severity grows. And the experimental results are similar.

Therefore, we could diagnose the cage friction fault by observing whether there are characteristic spectral lines of cage, rolling element and sidebands.

  1. In Conclusions you mentioned that tests were conducted under small load, but I can’t find the specified value for it.

Response:

When the flywheel bearing is working, the actual work is small load, and it is also small load in the experimental test. The specific load is mainly gravity.

  1. Referring to the experimental part, please describe the test rig as it can be very interesting point to the readers.

Response:

We have added the figure of the test rig as Fig.35(b).

  1. In the paper I miss the strict comparison between the model results and experiment.

Response:

Comparison 1: It can be seen from Figures 28, 30 and 32 that when the degree of friction fault is from weak to strong, the double sideband of cage is also from weak to strong. It can be seen from Figure 38 and Figure 41 that the same sideband amplitude rule exists.

Comparison 2: It can be seen from Figures 33 and 34 that the curve of modulation ratio of double sideband also changes from low to high as the friction fault changes from weak to strong. It can be seen from Figure 42 that when the friction fault is serious, the second-order sideband modulation ratio is much higher than that of minor fault.

  1. In Conclusions, please specify how do you want to extend your research.

Response:  We have added the description as:

In the future, we will establish a simulation model considering the detailed friction dynamic process to quantitatively evaluate the fault characteristics under different friction conditions, which is used to quantitatively evaluate the degree of friction fault.

The authors would like to thank the editors and reviewers for your valuable comments, which are extremely important for the improvement of this manuscript.

Thanks again.

Kind regards,

The authors

Round 2

Reviewer 1 Report

Good with the revision. 

Reviewer 2 Report

Dear Authors,

all the remarks have been introduced into the manuscript. I can recommend it for its publishing in its present form.

Best Regards

Reviewer